# Band structure engineered layered metals for low-loss plasmonics

Morten N. Gjerding[1,2], Mohnish Pandey[1] & Kristian S. Thygesen[1,2]

Plasmonics currently faces the problem of seemingly inevitable optical losses occurring in the metallic components that challenges the implementation of essentially any application. In this work, we show that Ohmic losses are reduced in certain layered metals, such as the transition metal dichalcogenide $TaS_2$, due to an extraordinarily small density of states for scattering in the near-IR originating from their special electronic band structure. On the basis of this observation, we propose a new class of band structure engineered van der Waals layered metals composed of hexagonal transition metal chalcogenide-halide layers with greatly suppressed intrinsic losses. Using first-principles calculations, we show that the suppression of optical losses lead to improved performance for thin-film waveguiding and transformation optics.

[1] Center for Atomic-scale Materials Design (CAMD), Department of Physics, Technical University of Denmark, Anker Engelundsvej 1, 2800 Kgs. Lyngby, Denmark. [2] Center for Nanostructured Graphene (CNG), Department of Physics, Technical University of Denmark, Anker Engelundsvej 1, 2800 Kgs. Lyngby, Denmark. Correspondence and requests for materials should be addressed to K.S.T. (email: thygesen@fysik.dtu.dk).

The basic idea of plasmonics is to utilise the sub-wavelength confinement of light on metallic surfaces in the form of plasmon polaritons to enable a range of applications including negative index materials[1], imaging[2–4], energy conversion[5,6], quantum information processing[7] and transformation optics (TO)[8,9]. However, in practice many of these applications are challenged by optical losses occuring in the metallic components[10–12]. So far, no viable alternatives to the noble metals have been found and the detrimental losses have appeared to be an unavoidable property of any conducting material.

Optical losses in metals originate from electronic transitions between the occupied and unoccupied parts of the crystal band structure (Fig. 1). In the absence of any scattering mechanisms, absorption of photons can take place only via vertical transitions in the Brillouin zone due to the small momentum of the photon, cf. transition (iii). However, scattering on crystal imperfections, phonons or other electrons, can provide the extra momentum required for light absorption via non-vertical electronic transitions, cf. transitions (i) and (ii). Typically, losses increase significantly above the onset of interband transitions which sets a hard upper limit on the operating frequencies of a plasmonic material. The effect of intraband scattering (transition (ii)) on the optical properties of metals is often accounted for by a phenomenological relaxation time. As we show here, a key to describe and discover metals with low intrinsic losses is to move beyond the commonly used constant relaxation-time approximation and include band-structure effects in the scattering rate.

Previous searches for new plasmonic materials in the optical regime have explored the alkali-noble intermetallics[13,14] and transition metal nitrides[15]. Some materials showed promise although their performance was still lower than that of the noble metals. Because all optical losses ultimately depend on the existence of an initial occupied and final unoccupied electronic state, one strategy for reducing losses has been to reduce the number of states available for scattering. Following this principle, doped semiconductors and transparent conducting oxides have been proposed for applications in the mid and near infrared (IR), respectively[16]. Recently, graphene plasmonics has been extensively studied due to its low density of states (DOS) around the Dirac cone which reduces losses and its easily tuneable plasma frequency[17,18].

The elusive loss-less metal[19] represents the ideal plasmonic material where no states are available for scattering in a particular frequency range thus eliminating optical losses completely, cf. transition (iv) in Fig. 1. Such a metal is obtained when the metallic, that is, partially filled, bands are separated from all higher and lower lying bands by sufficiently large energy gaps. Metals with such isolated intermediate bands are very rare among the conventional bulk materials. On the other hand, low-dimensional materials, in particular, layered materials, with a significant fraction of under-coordinated atoms and thus overall weaker hybridization, might be more likely to exhibit such characteristic band structures.

In this paper we test this hypothesis by examining the computed band structures of the metallic layered materials identified in ref. 20. We discover that the transition metal dichalcogenides (TMDs) $TaS_2$ and $NbS_2$ as well as aluminium(II) chloride, $AlCl_2$, have monolayer band structures that resemble that of the elusive loss-less metal, and confirm by first-principles density functional theory (DFT) calculations that the special electronic structure does entail lowered optical losses. The constant relaxation-time approximation is shown to be insufficient for describing the optical permittivity of these materials, and we propose a simple model of the scattering rate based on the joint DOS to remedy this deficiency. On the basis of

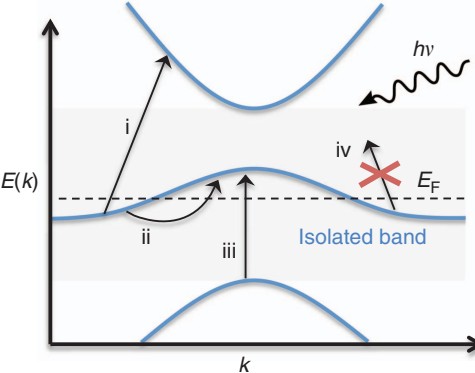

**Figure 1 | Origin of optical losses in metals.** The optical loss in metals originate from electronic transitions from occupied to unoccupied states in the crystal band structure (i, ii, iii). Only vertical transitions (iii) are allowed if scattering mechanisms are disregarded. Scattering mechanisms (impurity scattering, phonon absorption and emission, electron–electron scattering) can give the momentum required to allow for indirect transitions (i, ii). A metal with an intermediate band separated by gaps will suppress the number of transitions available (direct and indirect) at specific frequencies like in (iv) where no unoccupied state exists. When the gaps are sufficiently large to separate intraband (i) from interband (ii, iii) losses, some spectral ranges become immune to all quasi-elastic loss mechanisms.

the promising results obtained for the TMDs, we propose a new class of band structure engineered layered materials composed of transition metal chalcogenide-halide layers that greatly suppress the optical losses, and demonstrate the improved performance of these materials for thin-film waveguiding and TO.

## Results

**Identifying low-loss layered metals.** Examining the computed band structures of the metallic layered materials identified in ref. 20 by data filtering the Inorganic Crystal Structure Database (ICSD, Supplementary Table 1) reveal that the TMDs $2H\text{-}TaS_2$ and $2H\text{-}NbS_2$ as well as aluminium(II) chloride, $AlCl_2$, have monolayer band structures that resemble that of the elusive loss-less metal. While $2H\text{-}TaS_2$ and $3R\text{-}NbS_2$ have been synthesized in bulk form, $AlCl_2$ was derived from the experimentally known $AlCl_3$ by removing an interstitial Cl layer.

Figure 2 shows the atomic structures, band structures and DOS for (a) $2H\text{-}TaS_2$, (b) $3R\text{-}NbS_2$ and (c) $1T\text{-}AlCl_2$. The band structures of the monolayers are shown (red), although we will focus on the bulk structures throughout this work. In the case of $2H\text{-}TaS_2$ and $3R\text{-}NbS_2$, the formation of the bulk from the monolayer breaks the degeneracy of the highest valence band at the $\Gamma$ point and closes the gap to the metallic band, but the DOS above and below the intermediate band remains fairly low. Interlayer hybridization plays a smaller role for $1T\text{-}AlCl_2$; in particular, the band gaps are present for both the monolayer and bulk. The metallic band in $AlCl_2$ is formed by the Al s-states, and the small band width of 1 eV is a consequence of the weak hybridization between the Al atoms within the same atomic plane with the Cl atoms acting mainly as spacers. We stress that $AlCl_2$ is thermodynamically unstable. Although its calculated heat of formation is negative ($-0.89$ eV per atom) relative to the standard states of Al and Cl, it lies 0.7 eV above the convex hull when considering other competing phases (see Methods). In contrast, the layered compound $AlCl_3$ is stable and has been experimentally synthesized. Starting from $AlCl_3$, it may be feasible to remove the loosely bound Cl interstitial layer and thereby form metastable $AlCl_2$.

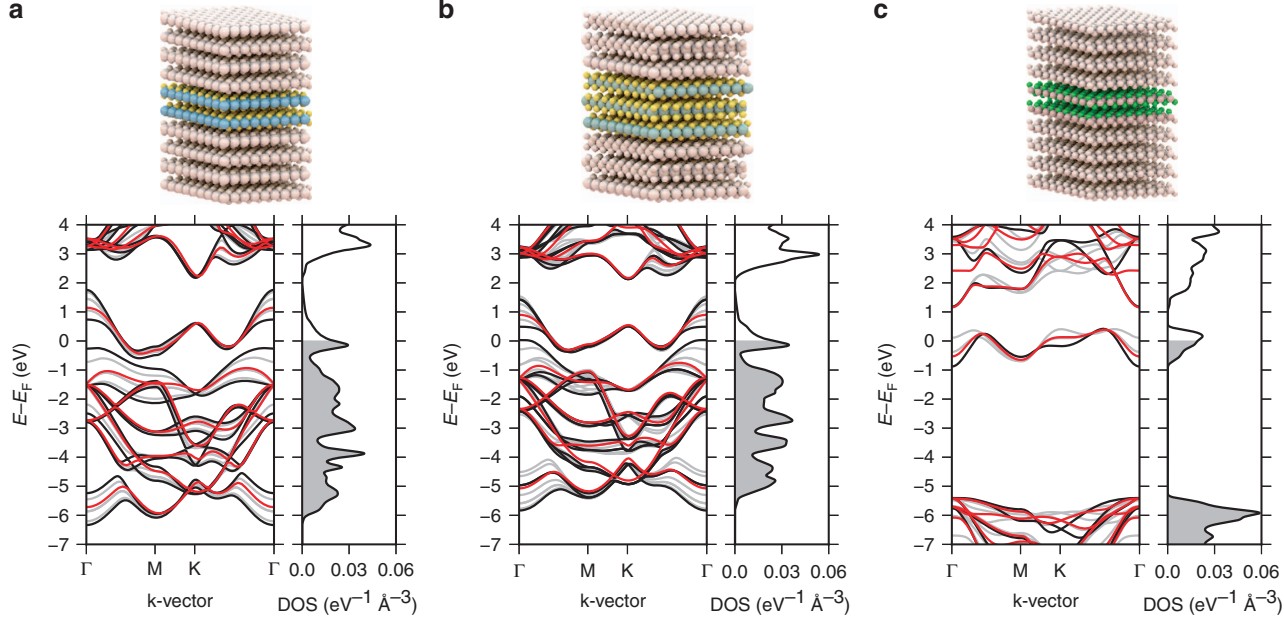

**Figure 2 | Band structures of vdW metals.** (**a**) 2H(AA′)-TaS$_2$, (**b**) 3R-NbS$_2$ and (**c**) 1T-AlCl$_2$ exhibit special band structures where the metallic band is separated from higher and lower lying bands by finite energy gaps. Such band structures entail a strongly reduced number of states available for scattering as depicted in Fig. 1. Comparing the monolayer band structures (red lines) with the bulk band structures (black) reveal that interlayer hybridization break the valence band degeneracy at the Γ point for 2H-TaS$_2$ and 3R-NbS$_2$, and closes the gap below the Fermi level. The interlayer hybridization is non-essential for 1T-AlCl$_2$. The dispersion in the out-of-plane direction does not effect the existence of an intermediate band as evidenced by the grey lines showing band structures at band paths translated in the out-of-plane direction.

We verify that the gapped metallic band structures entail lowered optical losses by performing first-principles calculations to obtain the dielectric function of AlCl$_2$ with an Al vacancy introduced in a $3 \times 3 \times 2$ supercell. The introduction of a vacancy is necessary to probe the optical losses due to intraband transitions where the defect provides the required momentum, see Fig. 1, transition ii. (Qualitatively similar results would be expected from phonon or electron scattering, however, such processes are more complicated to describe from first principles.) For comparison, a similar calculation was performed for a $3 \times 3 \times 3$ supercell of Ag with a single vacancy (see Methods). Figure 3 (full lines) shows the imaginary part of the dielectric function which is directly related to the optical losses of the metal. Optical losses at low frequencies due to the intraband transitions are indeed observed.

The dielectric function of a metal can be divided into contributions from interband and intraband transitions. It is customary to approximate the latter by a Drude response:

$$\varepsilon(\omega) = \varepsilon_{\text{inter}}(\omega) + \varepsilon_{\text{Drude}}(\omega), \qquad (1)$$

$$\varepsilon_{\text{Drude}}(\omega) = 1 - \frac{\omega_{\text{p}}^2}{\omega^2 - i\omega\eta}, \qquad (2)$$

where $\omega_{\text{p}}$ is the plasma frequency and $\eta$ is the constant relaxation rate of the free carriers. In Fig. 3, we show the result of the Drude model (thick grey lines) with $\eta$ fitted to obtain the best correspondence with the first-principles results using the bulk plasma frequency for $\omega_{\text{p}}$. Excellent agreement is found for silver all the way up to the onset of interband transitions demonstrating the validity of the constant relaxation-time approximation for this material. In contrast, it is not possible to fit the dielectric function of AlCl$_2$ by the simple Drude model below the onset of interband transitions at 1.5 eV. This is a result of the special band structure of AlCl$_2$, which introduces a strong energy dependence of the joint DOS (JDOS). The JDOS gives the density of electron–hole pairs at a given frequency (disregarding momentum

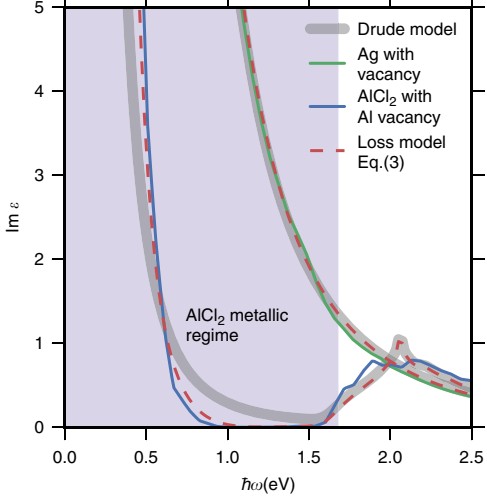

**Figure 3 | Defect induced optical losses and energy-dependent relaxation-time model.** The calculated optical losses (Im$\epsilon$) for silver (green) and AlCl$_2$ (blue) with a vacancy defect. The thick grey line shows the standard Drude expression (2) with a constant relaxation time, $\tau = 1/\eta$, fitted to the ab-initio results. For silver this provides an accurate description of the low-frequency (intraband) losses while the description is less accurate for AlCl$_2$ where it misses the strong suppression of losses in the 1–1.5 eV range. Results of the Drude expression with the energy-dependent relaxation rate model (3) (dashed lines) are in excellent agreement with the ab-initio results for both systems.

conservation) and thus represents the density of final states entering the expression for the scattering rate. In particular, the Drude model is not able to capture the strong reduction of the losses in the range 1–1.5 eV. In this energy range there are essentially no states available for scattering because of (i) the finite band width of the partially filled metallic band and (ii) the

presence of band gaps separating the metallic band from the rest of the bands. Consequently, the JDOS will vanish at energies above the intraband transitions and below the onset of interband transitions. In the case of $AlCl_2$, the band gaps are not large enough to completely separate the interband transitions from the intraband transitions, and consequently neither the JDOS nor $Im\epsilon$ become exactly zero. However, the JDOS is strongly suppressed in the range 1–1.5 eV and this is the origin of the low-loss nature of $AlCl_2$.

**Relaxation rate model.** A more realistic description of the relaxation rate should include information about the number of available electron–hole transitions, that is, the JDOS. A simple way to incorporate such a dependence is via

$$\eta(\omega) = a\frac{JDOS(\omega)}{\omega} \qquad (3)$$

where $a$ is a constant controlling the scattering strength. The expression can be motivated in different ways, but here it suffices to observe that (i) it reduces to the constant relaxation-time approximation for a metal with a constant DOS in a region around Fermi energy (in which case the $JDOS \propto \omega$) and (ii) it correctly leads to a vanishing relaxation rate for energies where there are no final states for the scattering. As can be seen from Fig. 3 (dashed lines) the model provides an excellent description of the optical losses of both materials. In the rest of the paper we use equation (3) to include scattering effects in a generic way setting the constant $a$ to the same value for all materials. This value is chosen to reproduce the experimental line width of the Ag surface plasmon (Fig. 4c).

We note that the JDOS model accounts only for first-order (quasi-)elastic scattering processes that generate single electron–hole pairs. Processes that generate multiple electron–hole pairs, higher-order processes or first-order processes such as the electron–electron scattering, could lead to absorption (loss) at energies between the intraband and interband transitions, where the JDOS model would otherwise predict $Im\epsilon$ to vanish. We expect losses due to higher-order processes to be small in the near-IR and losses due to first-order processes that involve the generation of multiple electron–hole pairs to be small due to the associated small phase space volume of final states.

The potential of the van der Waals (vdW) metals has been evaluated for two different plasmonic applications: Plasmonic waveguiding and TO[21] discussed in detail below. The calculated dielectric functions, DOS, JDOS and relaxation rate $\eta$ for all the layered metals studied in this work are provided in Supplementary Figs 9–29.

**Plasmonic waveguiding.** Plasmonic waveguides are envisioned as faster and less power consuming alternatives to the electronic interconnects currently used in integrated circuits. Essential prerequisites for such applications are long plasmon lifetimes and propagation lengths as well as tight spatial confinement of the plasmon to interface efficiently with the nanometre scale electronic components[22]. To evaluate the potential of the layered metals for such applications, we investigate plasmon propagation on single interface waveguides and on 5 nm thin-film waveguides, respectively (Fig. 4a,e). $SiO_2$ was used as the surrounding dielectric with a dielectric constant of $\varepsilon = 2.1$ ($\lambda \sim 700$ nm).

Figure 4 shows the normalized propagation length, $Re(k_p)/Im(k_p)$, where $k_p$ is the plasmon wave vector (b + c), the plasmon lifetime (c + g) and the mode extension into the dielectric (d + h), for silver, $2H-TaS_2$, $1T-AlCl_2$ and the chalco-halide HfBrS, to be discussed later. The plasmon lifetime, $\tau$, is calculated from the relationship $L = v_p\tau$, where $v_p$ is the plasmon velocity

(see Methods for details). The scattering strength ($a$) determining the frequency-dependent relaxation rate in equation (3) has been fixed by fitting the experimental result for the lifetime of the Ag surface plasmon.

For the thin-film waveguide, only the symmetric plasmon mode was considered. The light-like anti-symmetric mode extends far into the dielectric with mode widths on the order of 1 μm in which case the requirement of small coupling to the environment is not fulfilled. For the single interface plasmonic waveguide, silver shows significantly longer propagation lengths and lifetimes than any of the vdW metals. This is caused by the more efficient dielectric screening of Ag compared to the vdW metals, which implies that only a tiny fraction of the electromagnetic energy is contained within the silver surface (Supplementary Fig. 1a). The downside of the strong screening is the accompanying large mode widths caused by the electric field being pushed into the dielectric.

As the thickness of the thin-film plasmonic waveguide is reduced a larger fraction of the electric field is contained in the metal (Supplementary Fig. 1b) resulting in larger losses and decreasing plasmon group velocities[23]. Together, these two effects combine to give propagation lengths and lifetimes for the layered metals that are comparable to the noble metals. The dependence of the plasmon lifetime on the intrinsic optical losses and the electric-field distribution inside the metal is

$$\tau^{-1} \propto \int d\mathbf{r}|E_0(\mathbf{r})|^2 Im\varepsilon. \qquad (4)$$

From this it is clear that the relatively better performance of the layered metals in thin-film waveguides as compared to Ag must be due to lower intrinsic losses because the electric-field fraction within the vdW metal is much larger than in silver (Supplementary Fig. 1b).

It is well known that the PBE exchange-correlation (xc) functional underestimates both band gaps of semiconductors and interband transition energies in metals[24]. To test the influence of the xc-functional, we have computed the dielectric function of $2H-TaS_2$ using the range separated hybrid functional (HSE06), which generally yield more reliable band energies[25]. The primary effect of the HSE is to increase the size of the two gaps around the intermediate metallic band by around 0.5 eV (Supplementary Fig. 2) compared to PBE. This blue shifts the onset of interband transitions resulting in a significantly larger plasma frequency and lower losses. Consequently, the calculated plasmon lifetimes become superior to the noble metals while maintaining a small mode width (Fig. 4f–h).

**Band structure engineered layered metals.** With the goal of identifying new materials with band structures similar to those of $2H-TaS_2$ and $2H-NbS_2$, we have considered the class of layered materials of the form 2H-MXY (Fig. 5a), where M is a group 3 transition metal (Ti, Zr, Hf), X is oxygen or a chalcogen (O, S, Se) and Y is a halogen (Cl, Br, I). While conserving the total electron number, the lower electronegativity of the group 3 compared to group 4 transition metals, and the higher electronegativity of the halides compared to chalcogens, should increase the gap between the bands below and above the intermediate metallic band, which are composed mainly of chalcogen p and metal d, respectively. This should blue shift the interband transitions and decrease optical losses.

The calculated formation energies of the MXY compounds range from − 2.53 to − 0.92 eV per atom relative to the standard states. To check thermodynamic stability against other competing phases, we compare the formation energies to the convex hull (Supplementary Fig. 3; Supplementary Table 2). Out of the

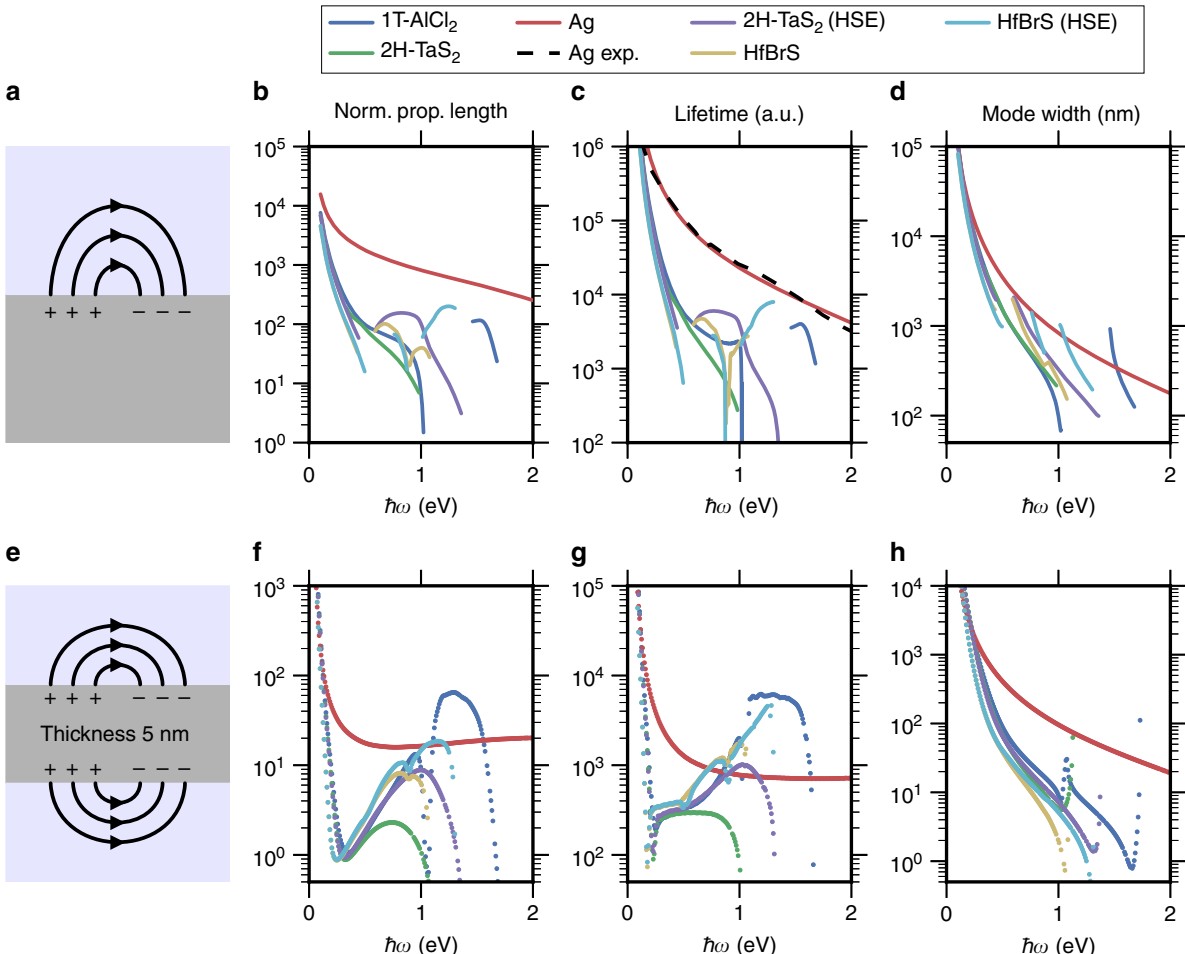

**Figure 4 | Plasmon waveguiding figures of merit.** The upper panels show the normalized plasmon propagation lengths $\mathrm{Re}(k_p)/\mathrm{Im}(k_p)$ (**b**), lifetimes (**c**) and mode width (**d**) for a plasmon on a single interface (**a**). The lower panels (**e**–**h**) show the same results for a 5 nm thin film. The more efficient dielectric screening in silver reduces the fraction of the electric field located within the metal as compared to the case of the vdW metals, resulting in lower losses, longer propagation lengths and lifetimes. On the other hand, since most of the electromagnetic energy is stored in the part of the electric field distributed over the dielectric, the mode widths of the surface plasmon becomes relatively large. Reducing the thickness of the metal to a 5 nm thin film pushes a larger fraction of the electric field into the metal leading to more confined modes. The redistribution of the electric field from dielectric to metal is more significant for silver explaining the overall better performance of the vdW metals for the thin-film geometry. Experimental data for the Ag surface plasmon was adapted from ref. 38.

27 compounds, 19 are found to be stable while the rest are only weakly unstable with formation energies lying <50 meV above the convex hull. The structural stability of the chalco-halides was checked by performing relaxations of the monolayers in super-cells containing $2 \times 2$ primitive cells (containing 12 atoms). With initial random distortions, the atoms relaxed back to restore the symmetry of the hexagonal lattice indicating that the structures are at least locally stable. The DFT calculations were performed fully spin polarized, but all the materials converged to a non-magnetic ground state.

The calculated $\mathrm{Im}\varepsilon$ are presented in Fig. 5b, and are characterized by significantly lower optical losses in the near-infrared (NIR) regime 1–1.5 eV; a direct signature of the reduction in the DOS for scattering (Supplementary Fig. 4). Figure 4 shows that the lowered optical losses entail an improved plasmon propagation length and lifetime as compared to 2H-TaS$_2$ for one of the compounds that exhibit the lowest optical losses, 2H-HfBrS, in both the single interface and thin-film geometries. As we found for 2H-TaS$_2$, the HSE functional increases the band gaps of the MXY compounds by $\sim 0.5$ eV which reduces the losses even further and increases the plasma frequencies, as illustrated for HfBrS (Fig. 5b). In the thin-film geometry, this

results in significantly improved plasmon lifetimes as compared to silver in the technologically important NIR frequency range (Fig. 4g) while maintaining a small mode volume (Fig. 4h).

**Transformation optics**. We now turn to an assessment of the potential of the layered metals for TO. Here the goal is to engineer the propagation of a wavefront through the volume of a metamaterial[21] for various applications. For a metamaterial structure containing metal-dielectric interfaces, it is required that the metallic response ($\mathrm{Re}\varepsilon_M$) is compensated by the dielectric response ($\mathrm{Re}\varepsilon_D$) while losses remaining low. Consequently, the typical figure of merit for TO applications is $\mathrm{FOM_{TO}} = 1/\mathrm{Im}\varepsilon_M$ while $-\mathrm{Re}\varepsilon_M \approx \mathrm{Re}\varepsilon_D$ (ref. 21). Assuming that the metallic response can always be matched by a dielectric within the range $1 < \mathrm{Re}\varepsilon_D < 20$, the relevant FOM becomes

$$\mathrm{FOM_{TO}} = \frac{1}{\mathrm{Im}\varepsilon_M}\theta(-\mathrm{Re}\varepsilon_M)\theta(\mathrm{Re}\varepsilon_M + 20) \qquad (5)$$

where $\theta(x)$ is the Heaviside step function. Figure 5c shows the calculated FOM for the MXY compounds and the best of the 18 metallic TMDs identified in ref. 20 (see Supplementary Table 1

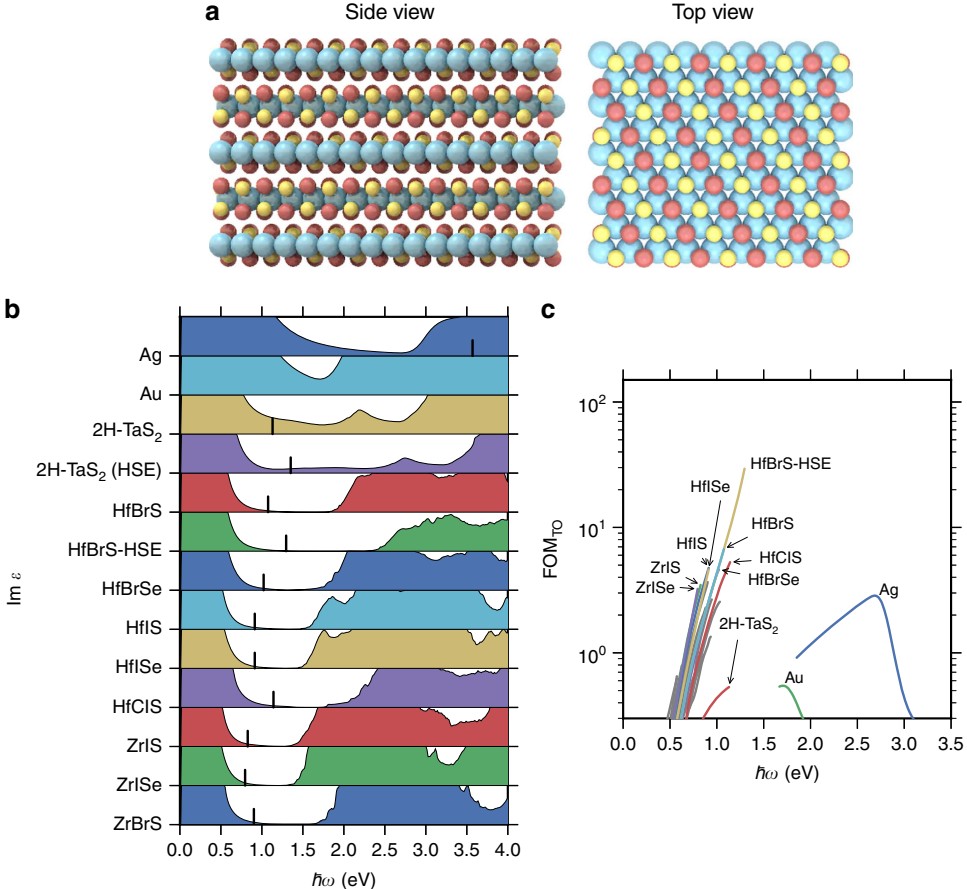

**Figure 5 | Optical properties of layer transition metal chalcogen-halogens.** Layered materials of the form 2H-MXY (**a**) where M is a group 3 transition metal, X is oxygen or a chalcogen (S, Se) and Y is a halogen (Cl, Br, I) are proposed as new plasmonic metals. The optical losses (**b**) shows that many of these compounds have considerable spectral ranges of low optical losses both compared to silver and to 2H-TaS$_2$ (the losses are normalized so that they can be directly compared between different materials, and the black bars denote the plasma frequencies.). This is due to the decreased electronegativity of (Hf, Zr, Ti) compared to Ta which increases the gap below the Fermi level. As a result, the predicted FOM (**c**) is orders of magnitude better than silver in the near-IR.

for the full list of materials). Reassuringly the best TMD turns out to be 2H-TaS$_2$ which is one of the metals with an intermediate metallic band. In general, the losses in the TMDs increase from sulfides to selenides to tellurides due to a lowering of the onset of interband transition. A similar trend is known for the semi-conducting TMDs where the band gaps decrease when progressing through the chalcogenide group towards higher atomic number[26]. All the MXY compounds outperform both 2H-TaS$_2$ and the noble metals in the technologically important near-IR frequency range. The improved performance is attributed to the larger gaps surrounding the intermediate metallic band which increases the plasma frequency and extends the loss-less regime between the intraband and interband transitions, see Fig. 5b.

**Loss-less layered halides.** Motivated by the excellent plasmonic properties of the (thermodynamically unstable) 1T-AlCl$_2$, we have performed a systematic study of layered bulk halides of the form (1T, 2H)-AY$_2$ where A is a group 3–13 element and Y is a halogen (Cl, Br or I). After filtering out materials with positive formation energies and finite magnetic moments and examining the band structures of the remaining materials, we identify several candidates with close to perfect resemblance with the loss-less metal. For example, 1T-GaCl$_2$ has an isolated intermediate metallic band of width 1.0 eV, an interband edge of 2.6 eV

(Supplementary Fig. 5a) and a plasma frequency of 1.4 eV (Supplementary Fig. 5b). As a result, the calculated plasmon propagation length and FOM$_{TO}$ are essentially infinite (Supplementary Fig. 5c). Unfortunately, these halides are unlikely to be thermodynamically stable when considering other competing phases, although we have not explored this systematically. Nevertheless, we find it interesting that these metastable structures exhibit such extreme properties.

## Discussion

A critical issue for the layered metals considered in this work is stability. Even though most of the chalco-halides were found to be thermodynamically stable relative to other competing phases involving the same elements, the materials could oxidize and disintegrate when exposed to air or moisture. Whether this will happen is largely a matter of kinetics, and a theoretical assessment requires calculation of reaction barriers which is beyond the present study. However, we note that encapsulation could be a way to protect the materials from reacting with other species. For example, it was recently demonstrated that encapsulation in hexagonal boron-nitride can protect and stabilize phosphorene in air, which in its free form is highly unstable and oxidizes quickly[27]. Airtight encapsulation of sensitive materials are routinely used in other fields such as organic light-emitting diodes where the organic layer is sealed between glass plates.

A potential problem that pervades all narrow band metals is the risk of a phase-transition to magnetic-, charge density wave- or Mott-insulating ground states. Such phase transitions are often driven by a high DOS at the Fermi level. Thus, the ideal metallic band for plasmonics should be wide enough to avoid such phase transitions and narrow enough to separate the intraband transitions from the interband transitions. Our calculations indicate that the chalco-halides are stable towards local structural distortions, but do not rule out the possibility of charge density wave-like distortions with larger periodicity. In fact, some of the metallic TMDs are known to exhibit charge density waves at low temperatures. On the other hand, at room temperature the presence of these phases is not expected to influence the optical properties that are governed by the gross features of the electronic structure. Moreover, the band width of the chalco-halides are comparable to those of the TMDs (Fig. 5b) which are known to behave as normal metals at room temperature, even for few layer samples[28].

Finally, we point to an added benefit of the vdW metals proposed in this work. Namely, their 2D nature allow them to be stacked with other non-metallic 2D materials like $MoS_2$ or hexagonal boron-nitride, to produce van der Waals metamaterials with ultra thin layer thickness and atomically well defined interfaces. We stress that it is very likely that the van der Waals metals proposed here should exhibit a lower scattering strength (controlled by the parameter $a$ in equation (3)) compared to conventional metals like silver and gold. Indeed, the noble metals primarily used in plasmonics, exhibit relatively high losses arising from intraband transitions via scattering on surface and interface roughness[11,29–31]. In contrast, the 2D materials form highly crystalline interfaces thus lowering the concentration of interfacial defects in actual devices. Combined with their intrinsic low-loss nature, this opens new perspectives for the fields of plasmonics and optical metamaterials.

## Methods

**Electronic structure code.** All electronic structure calculations have been performed with GPAW[32,33] employing plane wave basis sets and, unless explicitly stated, the PBE xc-functional. The linear response optical properties were calculated on top of PBE ground states using the random phase approximation (RPA) for the density response function. The $k$-point integration required in the calculation of the response function was performed using the linear tetrahedron interpolation scheme[34]. Local field effects were included in all calculations of the optical response but were found to be unimportant. Spin–orbit coupling is shown in Supplementary Fig. 6 to be negligible.

**Band structures.** The band structures of 2H-TaS₂ (ICSD ID: 651092), 3R-NbS₂ (ICSD ID: 645309) and 1T-AlCl₂ (AlCl₃ ICSD ID: 155670) were calculated with a plane wave cutoff of 600 eV and a $k$-point density of 10 Å$^{-1}$. In case of 1T-AlCl₂ (derived from AlCl₃), we determined the out-of-plane lattice constant to be 5.9 Å from structural relaxations using the optB88-vdW functional.

**Relaxation of layered compounds.** The out-of-plane lattice constant of all the layered MXY materials were determined using the optB88-vdW functional. A plane wave cutoff of 600 eV was used when relaxing unless the compound contained oxygen in which case a cutoff of 900 eV was used. This approach was found to reproduce the experimental interlayer-binding distance of 2H-TaS₂ to within 1% (Supplementary Fig. 7). The relaxed out-of-plane lattice constant of all the MXY components are presented in Supplementary Fig. 8.

**Supercell calculations of defects.** Optical response calculations reported in Fig. 3 employed supercells of $3 \times 3 \times 3$ and $3 \times 3 \times 2$ primitive cells for silver and 1T-AlCl₂, respectively. For silver, a Monkhorst-Pack $k$-point sampling of $30 \times 30 \times 30$ was employed, and for 1T-AlCl₂ we used a sampling of size $(24 \times 24 \times 16)$. A plane wave cutoff of 400 eV was employed in both response calculations. States up to at least 50 eV above the Fermi energy were included in the sum over states. These parameters were found to be sufficient for convergence of the optical response.

**Optical response of known TMDs.** In calculating the optical response of the layered metals identified in ref. 20 (Fig. 4), we employ a plane wave cutoff of 600 eV and a Monkhorst-Pack $k$-point density of at least 30 Å$^{-1}$. States up to 40 eV above the Fermi energy were included in the sum over states. These settings were sufficient to converge the optical properties.

**Surface plasmon propagation.** Surface plasmons on uniaxial substrates have been analysed in ref. 35. They show that in the case of purely real dielectric functions, the single interface surface plasmon polaritons (Fig. 4) exist only if $\varepsilon_\parallel < 0$ and either $\varepsilon_\parallel \varepsilon_\perp > \varepsilon_D^2$ or $\varepsilon_\perp > \varepsilon_D$, where $\varepsilon_D$ is the dielectric function of the dielectric and $\varepsilon_\parallel$ and $\varepsilon_\perp$ are, respectively, the in-plane and out-of-plane component of the dielectric tensor of the uniaxial material. The curves in Fig. 4b–d are only shown when the real parts of the dielectric functions fulfil one of these requirements.

For the insulator-metal-insulator waveguide the surface plasmon modes have been determined using the method of ref. 36. For TM-polarized modes in uniaxial waveguides the dispersion relations to be solved become

$$L + : \varepsilon_{M,\parallel} k_{\perp,D} + \varepsilon_D k_{\perp,M} \tanh\left(\frac{-ik_{\perp,M}d}{2}\right) = 0 \qquad (6)$$

$$L - : \varepsilon_{M,\parallel} k_{\perp,D} + \varepsilon_D k_{\perp,M} \coth\left(\frac{-ik_{\perp,M}d}{2}\right) = 0 \qquad (7)$$

for the antisymmetric $(L+)$ and the symmetric $(L-)$ modes with respect to the tangential electric field. Here, $k_{\perp,M} = \sqrt{\varepsilon_\parallel \omega^2/c^2 - \varepsilon_\parallel k_\parallel^2/\varepsilon_\perp}$ and $k_{\perp,D} = \sqrt{\varepsilon_D \omega^2/c^2 - k_\parallel^2}$ are the wave vector components perpendicular to the metal-insulator interface in the metal and dielectric, respectively.

**HSE06 calculations.** The HSE06 calculations were performed non-self consistently on top of PBE. In case of 2H-TaS₂, the HSE06 calculation employed a gamma-centered Monkhorst-Pack $k$-point sampling of (22, 22, 6) and a plane wave cutoff of 600 eV. For HfBrS, we used a gamma-centered Monkhorst-Pack $k$-point sampling of (8, 8, 4) and a plane wave cutoff of 600 eV.

**HSE response calculations.** For 2H-TaS₂, we found that a scissor operator of 0.55 eV on both gaps reproduced the DOS calculated by HSE (Supplementary Fig. 2). For HfBrS we found that scissor operators of 0.7 and 0.5 eV on the lower and upper gaps, respectively, was sufficient to reproduce the DOS of HSE. We calculated the HSE response by applying the scissor operators to the PBE ground states using the PBE transition matrix elements.

**Stability of MXY compounds and halides.** The monolayer stability of the transition metal chalcogen-halogen (MXY) compounds was evaluated based on the convex hull of the competing phases. The competing phases were determined using the Open Quantum Materials Database (OQMD) and includes 2–4 different structures in addition to the elemental phases. The heat of formation of the MXY compounds and all the competing phases were calculated using the fitted elemental phase reference energies for the standard states[37]. A Monkhorst-Pack $k$-point sampling of (7, 7, 3) and a plane wave cutoff of 800 eV was employed and found to be sufficient for converging the total energy. The computed hull energies and heats-of-formation are presented in Supplementary Table 2 and Supplementary Fig. 3.

**Optical properties of MXY compounds.** The optical properties of the MXY compounds from which the FOM was determined (Fig. 5) were calculated using a plane wave cutoff of 600 eV, a gamma-centered Monkhorst-Pack $k$-point sampling of size (18, 18, 8) and 300 bands (100 occupied) that was found to be sufficient for converging the optical properties. The optical response of the MXY compounds is anisotropic so the presented dielectric functions shows the average of the in-plane coefficients. This does not give rise to essential differences when comparing to specific components of the dielectric tensor.

**Data availability.** The implementation of the linear tetrahedron interpolation method for the calculation of the linear response optical properties can be found in the public repository https://gitlab.com/mortengjerding/gpaw. Linear response optical properties of the MXY compounds are available from the corresponding author upon reasonable request. The authors declare that all other data supporting the findings of this study are available within the paper and its Supplementary Information files.

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

## Acknowledgements

We gratefully acknowledge the financial support from the Center for Nanostructured Graphene (Project No. DNRF103) financed by the Danish National Research Foundation. The project received funding from the European Unions Horizon 2020 research and innovation programme under Grant Agreement No. 676580 with The Novel Materials Discovery (NOMAD) Laboratory, a European Center of Excellence.

## Author contributions

M.N.G. performed calculations and the data analysis under the supervision of K.S.T. except for calculations of stabilities which were performed by M.P. The first draft of the paper was written by M.N.G.

## Additional information

**Competing interests:** The authors declare no competing financial interests.

**Publisher's note**: 

