## [Peer Review File · Nature Communications]

Reviewers' comments:

Reviewer #1 (Remarks to the Author):

The work is very interesting as it presents a way to reduce the ohmic losses in the metals in the finite range of frequencies. In this the work goes beyond Ref [19] in identifying specific class of material with favorable band parameters. The paper should definitely be published in Nature Communications provide the authors improve it according to the comments below

1. The authors should also in principle consider the losses inflicted by the absorption aided by the Umklapp electron-electron scattering. In this process the photon energy may be larger than the band width and still absorption can take place as two electrons are excited. I do not think this factor is huge in near IR (it is in the visible or UV) but still....at least the authors should comment
2. I disagree with the authors' choice of the figure of merit. Really, one can think of many other figures of merit, in terms of ratio of real to imaginary part of dielectric constant. I am also not sure that it is easy to find a dielectric material with infrared dielectric constant of 50! I think that the authors must include the plot of both real and imaginary parts of permittivity vs. wavelength (As in Fig S9) and also scattering constant dependence $\eta(\omega)$ and then the readers can easily make their own conclusions.
3. The authors should elaborate on growth of chemically unstable (under oxidized) compounds. Has it ever been accomplished (not necessarily for these particular materials) and how?

Reviewer #2 (Remarks to the Author):

The manuscript presents a theoretical study, based on the density functional theory, on the dielectric function of a series of two-dimensional van der Waals metals.

While this work may help people working in the field to guide the search for useful plasmonic structures, I believe that the model used can provide only orientative results, as many potentially important factors are not included in the calculation.

After all, if I am not mistaken, the optical conductivity of such extensively studied materials as Silver or Gold have not yet been obtained from first principles.

In the case of 2D systems, structural instabilities, magnetic transitions or transitions to Mott insulating phases may completely change the computed results. Moreover, in the infrared regime the influence of phonons in the optical conductivity must be taken into account. This may be a severe limiting factor: in the case of graphene the existence of an optical phonon at 0.2 eV severely limits the application of this material for plasmonics for wavelengths smaller than six microns. Carbon-carbon bonds are the strongest in nature, so the considered materials surely have optical phonons at even smaller wavelengths that could largely modify the computed $\text{Im}(\epsilon)$.

Additionally, 2D materials should have much smaller screening than noble metals. Thus the influence of charged impurities should be much larger in the former case, something not taken into account in the theory either.

Some additional comments are:

- In my opinion, the title is not appropriate: 2D van der Waals materials are already being currently explored for low-loss plasmonics, and there is a very extensive literature on the topic (to the extent that two monographic REVIEW articles have appeared this week).

- The figure of merit considered may be used in the field of transformation optics, but it is not the common one in plasmonics. In this field the figure of merit is usually taken as the number of plasmon wavelengths that the plasmon propagates before its amplitude decreases by $1/e$, i.e. $FOM = \text{Re}(k_p)/\text{Im}(k_p)$, where k_p is the plasmon wavevector.

- It is redundant to represent the $FOM=1/\text{Im}(\epsilon)$ chosen and $\text{Im}(\epsilon)$ in the same figure (as in Figure 4).

- As I said, the calculations may be useful to guide research in this field. But for this it would be useful to be able to extract the dielectric constant for the studied materials. The results are provided in Fig. S1, but in a very unfortunate way that can not be used.

To summarize, I do not think this work provides more than an orientation to future work, but that reality may be very different to what the used theory predicts. Thus I do not think this paper deserves publication in Nature Communications.

Reviewer #1:

The work is very interesting as it presents a way to reduce the ohmic losses in the metals in the finite range of frequencies. In this the work goes beyond Ref [19] in identifying specific class of material with favorable band parameters. The paper should definitely be published in Nature Communications provide the authors improve it according to the comments below

We thank the referee for the positive evaluation of our work. Below we provide our answers to the comments.

1. The authors should also in principle consider the losses inflicted by the absorption aided by the Umklapp electron-electron scattering. In this process the photon energy may be larger than the band width and still absorption can take place as two electrons are excited. I do not think this factor is huge in near IR (it is in the visible or UV) but still....at least the authors should comment

We agree that higher-order processes involving the generation of several electron-hole pairs (e.g. mediated by the Coulomb interaction) or phonons, could lead to absorption losses at energies between the intraband and interband where our JDOS model would predict the material to be loss less. As the referee points out one would expect contributions from such higher-order processes to be small (at least in the NIR considered here). We have inserted a paragraph where we comment this issue as suggested by the referee.

2. I disagree with the authors' choice of the figure of merit. Really, one can think of many other figures of merit, in terms of ratio of real to imaginary part of dielectric constant. I am also not sure that it is easy to find a dielectric material with infrared dielectric constant of 50! I think that the authors must include the plot of both real and imaginary parts of permittivity vs. wavelength (As in Fig S9) and also scattering constant dependence $\eta(\omega)$ and then the readers can easily make their own conclusions.

We agree that there are many possible figures of merit. We have now included plots showing the normalized propagation length, $\text{Re}(k_p)/\text{Im}(k_p)$, of surface plasmon polaritons on a semi-infinite surface and a 5 nm thin film. In addition to the propagation length we provide the plasmon life-time and width. These quantities are shown in Figure 4 for Ag and a number of layered materials. For the thin film geometry, the SPP of the best layered materials have comparable propagation lengths, longer life times, and significantly better confinement than the Ag SPP.

We agree that a dielectric with permittivity of 50 is unrealistic (this choice was made a bit arbitrarily as it does not influence the conclusions). We have changed the cut-off value to 20 in the new manuscript.

We now provide in the SI the imaginary and real parts of the permittivity together with the frequency dependent scattering rate for all the materials investigated.

3. The authors should elaborate on growth of chemically unstable (under oxidized) compounds. Has it ever been accomplished (not necessarily for these particular materials) and how?

Many of the proposed chalcogenides are found to be thermodynamically (and structurally) stable relative to other competing phases involving the same elements. However, the materials could of course oxidize in contact with air or water. Whether this will happen is largely a matter of kinetics and requires calculation of certain reaction barriers. We feel this is beyond the scope of the present work. However, we stress that encapsulation of metastable materials in e.g. hexagonal boron-nitride could be a way to protect the materials from reacting with other species. For example, this method has been successfully used to protect and stabilize phosphorene in air, which in its free form is highly unstable and oxidizes within a few minutes (one example is: Nature Communications 7, 10450 (2016), but there are many similar reports). The problem is very similar to the case of organic light emitting diodes (OLEDs). If these were not protected from air, they would function for only a few seconds.

We have extended the discussion of this issue in the manuscript.

Reviewer #2:

We thank the referee for the critical yet constructive comments. We believe some of the issues raised by the referee rely on a misunderstanding concerning the type of materials we investigate in our paper which are not 2D monolayers but layered bulks. We have changed the title of the paper and several other formulations throughout the paper, to make this completely clear. Following the referee's suggestions, we have made extensive revisions of the paper including new calculations of structural stability and figures of merit and expanded the SI providing the calculated permittivities of all the considered materials. Below we provide detailed answers to all the specific comments.

The manuscript presents a theoretical study, based on the density functional theory, on the dielectric function of a series of two-dimensional van der Waals metals.

While this work may help people working in the field to guide the search for useful plasmonic structures, I believe that the model used can provide only orientative results, as many potentially important factors are not included in the calculation.

After all, if I am not mistaken, the optical conductivity of such extensively studied materials as Silver or Gold have not yet been obtained from first principles.

This is not correct. The dispersion relation (energy vs momentum transfer) of both the conventional surface plasmons, and the acoustic surface plasmons of Cu, Ag, and Au calculated from first-principles using the same methods as we apply in the present work, are in excellent agreement with experiments (see e.g. PRB 86, 241404 (2012)).

Moreover, first-principles calculations were recently shown to reproduce accurately the experimentally measured linewidths of surface plasmon polaritons on noble metal surfaces (ACS Nano 10,957 (2016)). Our calculations for the SPP lifetime on a Ag surface are in good agreement with these results (see Figure 4 of the new manuscript). The crucial ingredients for this are a correct description of the onset interband transitions, Fermi velocities and the less trivial effects of impurity and phonon scattering. In ACS Nano 10,957 (2016) the latter were evaluated ab-initio using second order perturbation theory. In our work we use the semi-empirical "JDOS model" for the frequency dependent scattering rate to account for such processes. The strength of the scattering potential is a free parameter of this model, but the example of AlCl₃ with a S vacancy (Fig. 2) shows that it can reproduce the full ab-initio results with explicit inclusion of defect scattering by fitting the scattering strength. For a consistent treatment of the materials we use the same scattering strength for all materials. As such our work is not intended to provide quantitatively accurate descriptions of the optical conductivities, but to use semi-quantitative models to predict the merits of non-conventional metals for plasmonics. Obviously, more work is

required to understand and characterize their detailed optical properties, e.g. using the methods applied in the ACS paper mentioned above.

In the case of 2D systems, structural instabilities, magnetic transitions or transitions to Mott insulating phases may completely change the computed results.

First of all, we get the impression that the referee thinks our work concerns two-dimensional materials in monolayer form. We stress that this is not the case. All the materials we considered are **bulk** layered materials. We refer to these materials as 2D van der Waals crystals, but they are bulk materials. To avoid misunderstandings we have changed the formulation on several places from “2D materials” to “layered materials”

To test the structural stability we have performed structural relaxations using larger (2x2 in-plane) and find no deviations from the perfect crystal. Concerning magnetic transitions, all our calculations were performed with spin polarization, and only materials with zero magnetic moment were considered in the paper. Thus we believe that the materials are stable against structural and magnetic transitions. Concerning Mott insulator transitions, this is beyond the DFT description. For this reason we explicitly mention that this possibility exists. On the other hand the band width of the considered metals is around 2 eV, i.e. the electronic states forming the bands are not highly localized, which we believe is large enough to prevent such transitions at least at room temperature. In fact, many of the proposed chalcogen-halides have electronic structure very similar to TaS₂ and NbS₂ which are known to behave as normal metals at room temperature, see e.g. *Two-dimensional metallic NbS₂: growth, optical identification and transport properties*, 2D Materials, 3, 025027 (2016).

We have expanded the discussion of these stability related issues in the new version of the manuscript.

Moreover, in the infrared regime the influence of phonons in the optical conductivity must be taken into account. This may be a severe limiting factor: in the case of graphene the existence of an optical phonon at 0.2 eV severely limits the application of this material for plasmonics for wavelengths smaller than six microns. Carbon-carbon bonds are the strongest in nature, so the considered materials surely have optical phonons at even smaller wavelengths that could largely modify the computed $\text{Im}(\epsilon)$.

We are not sure we can follow the argument of the referee here. First, the phonons in the proposed materials are certainly lower in energy than the 0.2 eV optical phonons in graphene, and not higher in energy (smaller in wavelength) as suggested by the referee. The effect of phonon-scattering is accounted for to first order by the semi-empirical relaxation rate used in our calculations.

The referee might be referring to higher-order effects involving the generation of several phonons or electron-hole pairs in one coherent photon-mediated event. Such processes could indeed lead to absorption losses at energies between the

intraband and interband transitions where the JDOS model would otherwise predict the material to be loss less. However, we expect losses due to such higher-order processes to be small in the near-IR. Moreover, the calculation of response functions taking higher-order phonon processes into account would be a daunting task and we are not aware of previous first-principles work achieving that.

We have included a short description of the role of higher-order scattering processes in the new version of the manuscript.

Additionally, 2D materials should have much smaller screening than noble metals. Thus the influence of charged impurities should be much larger in the former case, something not taken into account in the theory either.

The materials we consider are bulk metals. Their free-electron plasma frequencies lie around 3 eV which is certainly smaller than most elementary metals (Na,Al,Ag,Au), but they would still screen charged impurities efficiently (exponential screening although slightly larger screening length than in “standard” metals). Again, it appears that the referee thinks we consider atomically thin materials.

Some additional comments are:

- In my opinion, the title is not appropriate: 2D van der Waals materials are already being currently explored for low-loss plasmonics, and there is a very extensive literature on the topic (to the extent that two monographic REVIEW articles have appeared this week).

We agree that the title is maybe not the best and we have changed it to “Band structure engineered layered metals for low-loss plasmonics” to stress that we consider bulk materials. We certainly acknowledge that 2D plasmonics is, and has for a few years, been an active research field, but cannot see why this should be a problem in this context. Moreover, our work does not address atomically thin 2D materials, but a completely new class of layered metals with special band structures that suppress the number of final states for scattering – something that has not been considered before within 2D plasmonics (or plasmonics in general for that matter). In fact the field of 2D plasmonics has been almost exclusively focusing on graphene and hBN. Our work opens up new possibilities within this field by focusing on different classes of 2D metals whose plasmonic and optical properties are completely unexplored.

- The figure of merit considered may be used in the field of transformation optics, but it is not the common one in plasmonics. In this field the figure of merit is usually taken as the number of plasmon wavelengths that the plasmon propagates before its amplitude decreases by $1/e$, i.e. $FOM = \text{Re}(k_p)/\text{Im}(k_p)$, where k_p is the plasmon wavevector.

We agree that there are many possible figures of merit. We have now included plots showing the normalized propagation length, $\text{Re}(k_p)/\text{Im}(k_p)$, of surface

plasmon polaritons on a semi-infinite surface and a 5 nm thin film. In addition to the propagation length we provide the plasmon life-time and field width. These quantities are shown in Figure 4 for Ag and a number of layered materials. For the thin film geometry, the SPP of the best layered materials have comparable propagation lengths, longer life times, and significantly better confinement than the Ag SPP.

- It is redundant to represent the $FOM=1/\text{Im}(\epsilon)$ chosen and $\text{Im}(\epsilon)$ in the same figure (as in Figure 4).

We have removed $\text{Im}(\epsilon)$ from the new FOM plot to avoid redundancy.

- As I said, the calculations may be useful to guide research in this field. But for this it would be useful to be able to extract the dielectric constant for the studied materials. The results are provided in Fig. S1, but in a very unfortunate way that can not be used.

We realize that it would be useful to provide as much data as possible. Thus we now present the imaginary and real parts of the dielectric function as well as the frequency dependent scattering rate from the JDOS model for all the material investigated. These plots are provided in the SI.

The authors did a decent job of answering the comments of BOTH reviewers (I carefully read answers to Referee 2 as well) and now the manuscript is almost ready to be published. I have one final comment

The authors write (line 190)

“We note that the JDOS model only accounts for
191 _first-order (quasi-)elastic scattering processes. Higher-
192 order processes involving generation of several phonons
193 or electron-hole pairs in one coherent photon-mediated
194 event, could lead to absorption (loss) at energies between
195 the intraband and interband transitions where the JDOS
196 model would otherwise predict Im_ω to vanish. We expect
197 losses due to higher-order processes to be small in the
198 near-IR and therefore neglect them in the following.

Actually the process in which two electron hole pairs are generated is the same order process as phonon assisted absorption. Phonon assisted absorption probability involves the product of matrix element of electron-photon interaction and the one for electron phonon interaction. The process involving generation of two electron hole pairs involves the product of the aforementioned matrix element of electron-photon interaction and the matrix element for electron-electron scattering. So, strictly speaking they are both second order processes. The difference is of course contained in the fact that the phase space (density of final states) for the two electron hole pair generation increases as frequency goes up (actually as ω^2). And herein lies the REAL reason why one can neglect this process. This process becomes important only at higher frequencies, generally above 2-3eV. So it would be appropriate if the authors just say this in the text.

I suggest that the authors consult a venerable book “Optical Properties of Solids” by F. Abeles where the story of electron-scattering-assisted absorption is explained in a very accessible form in Chapter 3, on Pages 109-110. More details can be found David Pines book “The Theory of Quantum Liquids”

The authors are of course correct about two-phonon processes – they are indeed the next order of perturbation and can be neglected easily.

With that, the paper would be ready for prime time

Reviewer #2 (Remarks to the Author):

I admit was confused by the name "2D van der Waals materials", as I had always seen this name in relation to atomically thin materials. Now, I am more convinced of the validity of the calculations for bulk materials.

Additionally, the presentation is now much better: data can be extracted, and the figure of merit is the one relevant to surface plasmon excitations.

Therefore, I recommend publication of the manuscript in Nature Communications.

The reviewer comments are in black and our answers in red.

The authors did a decent job of answering the comments of BOTH reviewers (I carefully read answers to Referee 2 as well) and now the manuscript is almost ready to be published. I have one final comment
The authors write (line 190)

“We note that the JDOS model only accounts for
191 _first-order (quasi-)elastic scattering processes. Higher-
192 order processes involving generation of several phonons
193 or electron-hole pairs in one coherent photon-mediated
194 event, could lead to absorption (loss) at energies between
195 the intraband and interband transitions where the JDOS
196 model would otherwise predict Im_ϵ to vanish. We expect
197 losses due to higher-order processes to be small in the
198 near-IR and therefore neglect them in the following.

Actually the process in which two electron hole pairs are generated is the same order process as phonon assisted absorption. Phonon assisted absorption probability involves the product of matrix element of electron-photon interaction and the one for electron phonon interaction. The process involving generation of two electron hole pairs involves the product of the aforementioned matrix element of electron-photon interaction and the matrix element for electron-electron scattering. So, strictly speaking they are both second order processes. The difference is of course contained in the fact that the phase space (density of final states) for the two electron hole pair generation increases as frequency goes up (actually as ω^2). And herein lies the REAL reason why one can neglect this process. This process becomes important only at higher frequencies, generally above 2-3eV. So it would be appropriate if the authors just say this in the text.

I suggest that the authors consult a venerable book “Optical Properties of Solids” by F. Abeles where the story of electron-scattering-assisted absorption is explained in a very accessible form in Chapter 3, on Pages 109-110. More details can be found David Pines book “The Theory of Quantum Liquids”
The authors are of course correct about two-phonon processes – they are indeed the next order of perturbation and can be neglected easily.

With that, the paper would be ready for prime time

We agree with the comment and therefore we have changed the paragraph as suggested to read:

“We note that the JDOS model accounts only for first-order (quasi-)elastic scattering processes that generate single electron-hole pairs. Processes that generate multiple electron-hole pairs, higher-order processes or first-order processes such as the electron-electron scattering, could lead to absorption (loss) at energies between the intraband and interband transitions where the JDOS model would otherwise predict Im_ϵ to vanish. We expect losses due to higher-order processes to be small in the near-IR and losses due to first-order processes that involve the generation of multiple electron-hole pairs to be small due to the associated small phase space volume of final states.”